# Regulator of G-Protein Signalling 9: A New Candidate Gene for Sweet Food Liking?

**DOI:** 10.3390/foods12091739

**Published:** 2023-04-22

**Authors:** Catherine Anna-Marie Graham, Beatrice Spedicati, Giulia Pelliccione, Paolo Gasparini, Maria Pina Concas

**Affiliations:** 1Cereneo Foundation, Center for Interdisciplinary Research (CEFIR), Seestrasse 18, 6354 Vitznau, Switzerland; 2Lake Lucerne Institute, Seestrasse 18, 6354 Vitznau, Switzerland; 3Department of Medicine, Surgery and Health Sciences, University of Trieste, 34149 Trieste, Italy; 4Institute for Maternal and Child Health—IRCCS, Burlo Garofolo, 34127 Trieste, Italy

**Keywords:** GWAS, *RGS9*, sweet food, food liking, body mass index, dietary behaviour, personalised nutrition, nutrigenomics, nutrigenetics, nutrition

## Abstract

Genetics plays an important role in individual differences in food liking, which influences food choices and health. Sweet food liking is a complex trait and has been associated with increased body mass index (BMI) and related comorbidities. This genome-wide association study (GWAS) aimed to investigate the genetics of sweet food liking using two adult discovery cohorts (*n* = 1109, *n* = 373) and an independent replication cohort (*n* = 1073). In addition, we tested the association of our strongest result on parameters related to behaviour (food adventurousness (FA) and reward dependence (RD) and health status (BMI and blood glucose). The results demonstrate a novel strong association between the Regulator of G-Protein Signalling 9 (*RGS9I)* gene, strongest single nucleotide polymorphism (SNP) rs58931966 (*p*-value 7.05 × 10^−9^ in the combined sample of discovery and replication), and sweet food liking, with the minor allele (A) being associated with a decreased sweet food liking. We also found that the A allele of the rs58931966 SNP was associated with decreased FA and RD, and increased BMI and blood glucose (*p*-values < 0.05). Differences were highlighted in sex-specific analysis on BMI and glucose. Our results highlight a novel genetic association with food liking and are indicative of genetic variation influencing the psychological–biological drivers of food preference. If confirmed in other studies, such genetic associations could allow a greater understanding of chronic disease management from both a habitual dietary intake and reward-related perspective.

## 1. Introduction

The importance of a healthy diet to prevent non-communicable diseases is well recognised [1,2,3]. For example, many studies underline the role of the intake of dietary pulses in the prevention of cardiometabolic diseases, such as obesity, hypertension, and cardiovascular diseases [1,2], and others have shown that total fruit consumption is inversely associated with cardiometabolic risk factors [3]. Conversely, over the past few decades, sugar intake has been a focal point for scientific research pertaining to its negative relationship with various non-communicable diseases [4,5]. Dietary intake is influenced by a multitude of factors, including both socio-demographics and genetics [6,7,8]. Food liking is the pleasure derived from the consumption of a food [4,9], determined by perceived flavour, and is thus a component of the overall palatability of a food [10]. Sweetness is an influencer of food liking [11,12]. This has been shown in both rodents [13], and, more recently, humans [14]. Multiple models have been described to define the hedonic responses to sweet foods. For example, in 1996, Tuorila [15] reported an inverted U-shaped response to sucrose concentration, denoting a general increase in liking with an increased sucrose concentration, followed by a decline when concentrations rise above the optimum. This classification is still used in practice today [16]. Others have categorised participants into distinct liker and disliker groups, denoting a monotonic increase in liking with a rising concentration for sweet likers, and a decrease or early plateau for sweet dislikers [17,18]. Alongside this, various other influencers of sweet liking have been described, including the age of exposure to sweet foods [19], ethnicity [20], association with other tastants [21], body mass; and, in more recent years, psychological state [22], including addiction [23] and individual differences based on genetic variation [24]. It is not yet clear the extent of influence that such factors may have on food liking.

To date, research regarding taste and genetics has focused on taste receptors genes [24,25,26,27]. The taste receptor type 1, member 2 and member 3 (*TAS1R2* and *R3*) genes are responsible for sweet perception [24] and might increase sweet food consumption or liking [28,29]. However, conflicting results have been reported [30], including the lack of association found with candidate genotypes in genome-wide association studies (GWAS). The most relevant of these studies was conducted by Hwang et al. [31], who did not find an association with the known taste receptor genotypes and sweet food liking but did report various other suggestive genetic associations. No similar research has been carried out in more recent years. Prior to this, in 2016, despite sweet liking not being assessed, Pirastu et al. [32] revealed fifteen independent genome-wide significant loci for the liking of twelve foods, assessed via Likert scale, within the categories of vegetables, fatty, dairy, and bitter. All things considered, it is evident that there is vast genetic heterogeneity in food liking and perception, with a strong impact on dietary intake. In this light, it is important to consider the wider consequential factors of food liking: for example, an increased high-energy-dense food diet variety, namely sweet and fatty foods, has been associated with body mass index (BMI) [33] and blood glucose level [34], while nutrient–gene interactions have been demonstrated with both [35,36]. However, not all foodstuffs may have this effect; an increased consumption of low-energy-dense, healthier foodstuffs may aid in losing weight, although results are unclear [33]. In addition, the reward system, specifically relating to dopamine signalling, considerably influences sweet food preference [37]. Sweet food consumption has been shown to elicit similar responses to opioids and other addictive substances, with genetic variability in the dopamine receptor (*DRD2*) influencing response extent [37]. Recently, other psychological influencers have been noted, although these are generally less defined [38]. For example, food adventurousness (FA), which has recently been demonstrated to influence whole diet variety [38,39], has previously been shown to influence the liking of more pungent and spicy foods within taster groups specifically regarding 6-n-propylthiouracil (PROP) taster status [40]. Taken together, research regarding sweet food liking and genetic variation should consider the relationship with BMI, blood glucose, and psychological influencers of food preference.

Here, we report the results of a GWAS carried out on the liking of sweet foods with the aim to both verify previously reported genetic associations and reveal new genetic variants, using two Italian populations as the discovery cohort and an independent Italian cohort as the replication cohort. In addition, we tested the association of our strongest result on parameters related to behaviour (FA, and reward dependence (RD)) and health status (BMI and blood glucose).

## 2. Materials and Methods

### 2.1. Participants

The discovery phase was carried out on a total of 1482 participants divided as follows: 1109 from the Val Borbera (VBI) cohort in Northern Italy, and 373 from Carlantino (CAR) in Southern Italy. The replication phase was performed on 1073 individuals from Friuli Venezia Giulia (FVG), a region in the Northeast of Italy. All three populations belong to the Italian Network of Genetic Isolates (INGI), a project aimed at studying the genetic basis of complex diseases and phenotypes (including eating habits) [41]. The ethical committees of San Raffaele Hospital and IRCCS Burlo Garofolo approved the study, and all participants signed their written informed consent.

### 2.2. Data Collection

Screening sessions were organised in each village in government-provided accommodation. Demographic, lifestyle information, and living habits were collected for each participant using a self-administered standard questionnaire. All questionnaires were carried out on the same day, after a detailed explanation by trained staff. Trained personnel were available at all times to answer participants’ questions.

To determine food liking in all populations, for approximately 100 foods and beverages, a 9-Point Likert scale was used: from “dislike extremely” (1) to “like extremely” (9) [32,42]. The mean liking of the following foods was used to establish opinion on overall sweet liking: ice cream, panettone, whipped cream, milk chocolate, marzipan, biscuits, cake, marmalade, Nutella, icing sugar, and hot chocolate with cream. The reliability of the group was >0.6 in each population (Cronbach alpha function available in R library psy).

Food adventurousness was assessed for each participant using the question “How often do you try unfamiliar foods?”. The response categories were: “never”, “rarely”, “some of the time”, “often”, and “very often”.

RD was assessed by the Temperament and Character Inventory (TCI) in FVG for a total of 587 participants [43].

Height (m) and weight (kg) were measured; then, BMI (kg/m^2^) was calculated. In VBI and FVG, height was measured to the nearest 0.25 cm using a stadiometer; then, weight and BMI were measured using the Body Composition Analyzer (Tanita BC-420MA; Tanita, Tokyo, Japan). In CAR, body weight was measured to the nearest 0.25 kg using a balance-beam scale, and height was measured to the nearest 0.25 cm using a stadiometer. BMI was manually calculated.

Fasting blood samples were obtained in separate sessions in the early morning. Blood was tested the same day or aliquoted and stored for further analysis. Routine biochemical analyses were performed through the Cobas 6000 analyzer (Hoffmann-La Roche, Basel, Switzerland).

### 2.3. Genotyping and Imputation

Genomic DNA was extracted from blood using a phenol–chloroform extraction procedure. All samples (discovery and replication) were genotyped with an Illumina 370/700 k high-density single nucleotide polymorphism (SNP) array. Genotype imputation was conducted after standard quality control (sample call rate ≥ 0.95, SNP call rate ≥ 0.99, Hardy Weinberg Equilibrium test *p*-value 1 × 10^−6^, Minimum Allele Frequency (MAF) 0.01) using IMPUTE2 [44]. The 1000 Genomes phase 3 [45] and whole genome sequences of the INGI samples were merged and used as a reference panel [41]. After imputation, SNPs with MAF < 0.01 and Info Score < 0.4 were discarded from the statistical analyses. The SNP position refers to the build hg19.

### 2.4. Genome-Wide Association Study and Meta-Analysis

GWAS were conducted in CAR and VBI separately, using mixed linear models (ABEL R packages) [46]. Genomic kinship matrices were used as random effects to consider relatedness. An additive model was used, and this was adjusted by sex, smoking (no/yes), intensity of perception of PROP, FA, cognitive restraint, and eating disinhibition. These covariates were chosen based on our previous work [47]. The meta-analysis was performed using an inverse-variance method (METAL) [48]. Variants with significant *p*-values (*p* < 0.05) for heterogeneity Cochran Q were excluded from the meta-analysis, and only concordant variants in the two populations were examined. SNPs with *p*-value < 5 × 10^−8^ were considered genome-wide significant. SNPs were annotated with the Variant Effect Predictor tool (https://www.ensembl.org/info/docs/tools/vep/index.html; accessed on 12 April 2022) [49], in order to determine the closest genes and to obtain functional characteristics. Linkage disequilibrium (LD) between SNPs was assessed in 926 whole genome sequences (424 from VBI, 124 from CAR, and 378 from FVG) using R library Genetics. The variance explained by the best SNP was calculated as the difference of the coefficient of determination (R^2^) in the full model (i.e., using as covariates sex, smoking, intensity of perception of PROP, FA, cognitive restraint, eating disinhibition, and SNP) and base model (only sex, smoking, intensity of perception of PROP, FA, cognitive restraint, and eating disinhibition).

### 2.5. Replication Analysis

In the replication phase, the GWAS significant SNPs were analysed using mixed linear models. Results from the discovery and replication steps were combined using an inverse-variance method (METAL) [48].

### 2.6. Genotype-Tissue Expression (GTEx) Analysis

An analysis of human gene expression levels was carried out using the GTEx Database, release v8 [50].

### 2.7. Association of the Strongest Result with Other Phenotypes

We tested the association of the strongest SNP with liking of the individual sweet foods and FA, BMI, and glucose in all the three populations. In addition, we tested the association of the strongest SNP with RD in FVG. Mixed regression models were performed to assess the relationship between tested variables considering the population as random effect (nlme package on R). To better understand the relationship between health parameters (BMI and glucose) and the SNP and sweet liking, we performed the analysis on the total sample and in males and females separately. To better understand the complex interplay between the variables, models were developed and tested with structural equation modelling (SEM, lavaan R package) [51,52], a statistical technique allowing the identification of direct and indirect influences of variables. SEM is used when the response variable in one regression equation becomes a predictor in another. Sex, age, and population (CAR, FVG, or VBI) were included in the models as potential confounders. The criteria for overall fit were chosen a priori: Chi-square *p*-value non-significant (Chi-square test, *p >* 0.05); Confirmatory Fit Index (CFI) ≥ 0.92; Tucker–Lewis Index (TLI) *>* 0.87; and root mean square error of approximation (RMSEA) *<* 0.05 [53]. To assess the significance of the mediated model, the Sobel test was used [54].

### 2.8. Association of the Sweet Liking Group with Known Genes/SNPs

Finally, we verified the association of the sweet liking group with taste-related genes and genetic variants already published, as associated with taste perception and food liking or intake in the combined samples of 2555 individuals. We considered as significant the nominal *p*-value of 0.05. All the analyses were performed using R (www.r-project.org, accessed 15 May 2022) v.4.0.0.

## 3. Results

The main characteristics of the analysed cohorts are summarized in Table 1, together with the sweet liking mean and the means of all the variables tested. The discovery populations showed similar gender distribution, while FVG showed a lower percentage of females compared VBI (Chi-square test *p*-value < 0.05). The mean age was higher in VBI compared to CAR and FVG (*p*-value < 0.05). No differences between populations were found in the sweet liking group distribution (*t*-test *p*-value > 0.05). FA was higher in VBI compared to FVG and CAR (Wilcoxon test *p*-value < 0.05), BMI was higher in CAR compared to VBI and FVG, while blood glucose was lower in VBI compared to the other two populations (*t*-test *p*-value < 0.05). Appendix A shows the mean and standard deviation of liking for the sweet foods included in the group.

### 3.1. GWAS and Meta-Analysis Results

Figure 1 shows the Manhattan plot of the results of the meta-analysis in the discovery populations, while Appendix A displays the QQ-plot. All the results with *p*-value < 1 × 10^−5^ are shown in Appendix A. Their variant effect predictor (VEP) annotation is available in Appendix A. The effect of population stratification was negligible, as confirmed by the values of the genomic inflation lambda = 0.9927 in the meta-analysis, and 0.9868 and 1.0046 in VBI and CAR GWAS, respectively.

Seventeen SNPs reached the statistically significant *p*-value < 5 × 10^−8^, and 16 of them were successfully replicated, as shown in Table 2. The strongest association in the discovery cohorts was found with the rs58931966 SNP on chromosome 17, and all sixteen replicated SNPs were from the (Regulator of G-Protein Signalling 9) *RGS9* gene region. Figure 2 shows the regional plot of the findings in the discovery cohorts. In this region, all the discovered SNPs are in linkage disequilibrium (LD) in all three cohorts (i.e., the minimum D’ was 0.98 in VBI, 0.997 in CAR, and 0.917 in FVG).

Due to the LD, only the SNP with the lowest *p*-value in the discovery phase (rs58931966, *p*-value 1.56 × 10^−8^, ~22 Kb near the gene) was selected for the following analysis. The minor A allele (MAF = 0.14) of this SNP is associated with a decrease in sweet liking of 0.37 (standard error 0.07) in the discovery cohorts and a decrease of 0.30 (standard error 0.05, *p*-value 7.05 × 10^−9^) in the combined samples of 2255 individuals (discovery and replication cohorts). The variance explained by the best SNP was 1.2%. Considering the individual food liking, we observed the same effect of the SNP: the A allele was associated with a significant (*p*-value < 0.05) decrease in liking for each sweet food, except marmalade (β −0.172, *p*-value 0.0843). The results are reported in Appendix A.

Using GTEx, we found that the *RGS9* gene is expressed in various tissues, but particularly in the brain (Figure 3a). Additionally, the rs58931966 SNP is a splicing quantitative trait locus (sQTL) for the same gene in pituitary tissue (*p*-value 9.0 × 10^−10^, NES (normalized effect size) = −0.53) (Figure 3b).

### 3.2. Association of RGS9 rs58931966 with Other Health and Eating-Related Psychological Traits

The A allele of the rs58931966 SNP was significantly associated (*p*-value < 0.05, Table 3) with lower levels of FA (β = −0.065) and RD (β = −3.840), and higher values of BMI (β = 0.391) and glucose (β = 1.211) (Table 3).

No statistically significant relationship between health parameters and sweet liking was found in the total sample. Linear mixed regression models with sex and age as covariates showed that sweet liking was not a predictor for BMI (β = 0.04, *p*-value 0.42) or glucose (β = −0.02, *p*-value 0.85). No influence of RD on sweet liking was observed (β = 0.001, *p*-value 0.93). Conversely, FA was a predictor for sweet liking (β = 0.12, *p*-value < 0.0001). Consequently, we evaluated the relationship between the SNP, sweet liking, and FA using SEM models and a Sobel test. The results showed that the relationship of the SNP with sweet liking was both direct and mediated by FA (Sobel test, z = 2.04, *p*-value = 0.041, Figure 4).

Performing a gender-specific analysis, we found that both the SNP and sweet liking were statistically significant predictors for BMI in the female sample (rs58931966 A allele, β 0.62, *p*-value 0.0013; sweet liking, β 0.15, *p*-value 0.0479). Regarding glucose, the association with the SNP was found only in males (A allele, β 1.76, *p*-value 0.0323), and no significant association with sweet liking was found. We did not find any relationship between FA/RD and BMI or glucose.

### 3.3. Association of the Sweet Liking Group with Other Known Genes/SNPs

We verified the association between previously published sweet taste liking and intake-related SNPs and the sweet liking group in the combined sample. The full list of tested SNPs and the results we found from the GWAS on the sweet liking group is available in Appendix A. Table 4 shows the significant results (*p*-value < 0.05).

## 4. Discussion

Here, we report a novel association between the *RGS9* gene and sweet food liking, with the minor allele (A) of the most associated SNP, rs58931966, being associated with a decreased sweet food liking. We also found an association of this gene with a decrease in FA and RD, and an increase in BMI and blood glucose levels. Our results present an association between genetics, food liking, dietary behaviour, and markers of chronic disease risk.

The *RGS9* gene codes for the RGS9 protein, which is a GTPase-accelerating protein and thus a negative regulator of G-protein signalling [57]. The RGS9 protein has two isoforms: RGS9-1 and RGS9-2. RGS9-1 is expressed in the retina and RGS9-2 is expressed mainly in the striatum and has been linked to reward systems, specifically dopaminergic signalling [58,59]. Due to the association found between rs58931966 and sweet food liking, and rs58931966 with psychological traits (FA and RD), we hypothesise that the rs58931966 SNP is involved in pathways related to RGS9-2. This finding is supported by *RGS9* being largely expressed in the brain and rs58931966 being a splicing quantitative trait locus (sQTL) for the same gene in pituitary tissue.

In keeping with our findings, Hwang et al. [31] reported thirteen suggestive genetic associations with sweet food liking. The SNPs associated with sweet liking in the study [31] did not reach the significant threshold of *p*-value < 0.05 in our sample; however, some SNPs were replicated that were associated with [31] the intake of total sugar and the perceived intensity of aspartame. Taken together, these results demonstrate that taste phenotypes are likely driven in part by a genetic element, although the extent of this remains unknown. This is applicable to the wider context of chronic disease research due to food liking driving dietary intake [60], with liking of more sweet, salty, and fatty foods being associated with chronic diseases such as type II diabetes mellitus [61], cardiovascular diseases [62], and some cancers [63]. Further, the contrasting genetic results between our manuscript and Hwang et al. [31] may be due to the ethnicity and cultural differences in the cohorts; Hwang et al. [31] assessed data from Australia and the US, whereas our cohort consisted of three isolated Italian populations. Despite this, the populations were of a similar sample size, although Hwang et al. [31] continued the analysis using a larger sample to explore dietary intake only (*n* = 174,424 White-British individuals, from the UK BioBank). We did not assess dietary intake; however, food liking is also used to capture habitual dietary consumption due to its reliance on affective memory as opposed to factual memory [64,65]. Interestingly, we replicated one SNP of *TAS1R2* and one of the *TAS1R3* genes associated with sweet perception [28,29,30]. In order to confirm and assess the applicability of the obtained results, further research is required on a larger sample size; despite this being the largest study of its kind, independent replication of the results is required.

Moreover, sweet, energy-dense foods are generally deemed palatable and are known to elicit dopaminergic signalling [66,67]. Functional magnetic resonance imaging (fMRI) has demonstrated that the consumption of high sugar foods leads to activation of the mesolimbic reward areas and gustatory regions of the brain [66,67]. In this context, we evaluated the role of *RGS9*, and FA and RD, demonstrating a possible relationship in an area in which there is great debate. Regarding FA, the A allele of rs58931966 was associated with a lower sweet taste liking and a lower FA; comparatively, a higher FA was associated with a higher sweet liking. We are the first to report a common genetic link in this area; however, others have reported a comparable association between FA and food liking. For example, Ullrich et al. [40] demonstrated, in 232 healthy American adults, that those who are more food adventurous tend to exhibit behaviours similar to non-tasters: an increased liking, specifically with more pungent, hot, or bitter foods. To date, no contrasting results have been published in this area; however, overall, few research studies have been carried out, making the results ungeneralisable, specifically in relation to ethnicity.

Further, in our sub-cohort analysis, we report that the A allele of rs58931966 was associated with a decreased RD; this is indicative of the liking of palatable foods and RD being somewhat controlled by genetics. The RSG9 protein negatively regulates, by accelerating the termination of, G protein signalling, which modifies reward response and is thus postulated to contribute to the development of addictive disease [68]. Rodent studies with *RGS9* knockout variants have demonstrated an increase in morphine-related reward [69]. Gene regulation patterns for food addiction and drug addiction are similar. Navandar et al. [59], using a gene ontology analysis, reported 56 upregulated genes in both palatable food and cocaine addictive rodents, including the *RGS9* gene. This evidence corroborates our findings, indicating that the *RGS9* gene may be involved in the reward-related pathways related to palatable foods, and thus supports its association with sweet food liking. The concept of sweet food liking being an addictive trait in humans is not new; several review articles both support and refute the notion [37,70,71,72,73]. Future research should endeavour to explore differences in reward-related signalling in humans when presented with palatable foods, based on the rs58931966 genotype, in order to determine the applicability of our findings in disease management.

Furthermore, an increased BMI and increased blood glucose, two key indicators of chronic disease risk, were associated with the *RGS9* gene. Waugh et al. [74] demonstrated in both rodents and human knockouts that a 5-nucleotide deletion polymorphism in the *RGS9* gene led to a significantly higher BMI (approximately 2.7% higher) when compared to those without. It was theorised that the deletion leads to a less functional protein, which may subsequently lead to a change in the activity of hypothalamic centres that control energy regulation, although the mechanism remains unknown. Walker et al. [75] report similar results using only rodents, as well as demonstrate gender differences whereby male KO mice accumulate a greater fat mass. Both works, and our own, are indicative of the *RGS9* gene being involved in energy regulation via alterations in reward-related brain signals. Although we did not find an association between FA or RD and BMI, the direction of associations reported with rs58931966 lends this further support. For example, more food adventurous individuals tend to consume a wider, healthier array of foods [40,76]. The majority of research in this area focuses on food neophobia in children, the opposite to FA (or food neophilia); few have translated findings to an adult cohort [33,40,77]. Lower neophobia in children has been associated with a higher Healthy Eating Index Score and a lower body weight [76,78], due to neophobia largely being associated with consuming a decrease in bitter foods, namely vegetables [76]. To our knowledge, only Latimer et al. [33] and ourselves have assessed the relationship between FA and body mass in an adult cohort. Latimer et al. [33] surveyed 501 US women, demonstrating that food adventurous eaters had a lower BMI when compared to non-adventurous eaters. We did not find this direct association, only a genetic variable associated with both individually, which was not assessed in Latimer et al. [33]. The cohort ethnicity, sample size, and method of assessment of FA differ between studies; thus, due to the limited research, the extent by which FA and its influencing factors impact BMI is unclear. However, considering our findings in combination with others [31], it can be hypothesised that genetic variation may mediate pathways interlinking the biological and psychological processes of eating behaviour.

Additionally, many have reported that an increased sweet food liking is associated with an increased BMI [33]; we found this in females only. However, to our knowledge, no published literature controls for FA in this context; thus, it can be hypothesised that those who are more food adventurous may like a wider variety of foods, including sweet foods, but their overall dietary consumption may not be biased towards sweet, high-energy-dense foods, which contrasts with the common associations made with sweet food liking and BMI. Further research should endeavour to explore whether eating behaviours, such as FA, can mediate overall dietary intake and, subsequently, BMI outcome, and if this is driven by genetic variation.

### Limitations 

Although the sample size is the largest of its kind, in comparison to other GWAS studies, it is relatively small; thus, a larger sample should be used to confirm the results [33]. In addition, the food liking data are self-reported and subjective. Self-reported data related to diet have limited accuracy and can be subject to bias. Additionally, food liking is routinely assessed by subjective measurement; objective measurements, for example, facial expression analysis, are in existence [77,78], but these have a higher participant burden and are not modelled for multiple emotions. Finally, our sweet food liking group was constructed with a limited number of foodstuffs; for results to be generalisable to different populations, a wider range of foodstuffs need to be included, or food groups should be considered.

## 5. Conclusions

In conclusion, our results describe a novel genetic association with food liking and are indicative of genetic variation influencing the psychological–biological drivers of food preference. If further researched, such genetic associations could allow a greater understanding of chronic disease management from both a habitual dietary intake and reward-related perspective. Specifically, GWAS, such as this manuscript, provide a basis for further research pertaining to the genes of relevance. As discussed, various genetic associations with sweet food liking, intake, and intensity, have been published; this, alongside the associations with eating behaviour and body mass, indicate that there is a potential strong role of genetics in food choice. However, in order to understand the extent of this relationship it is important to explore the role of *RGS9*, and other genes, in relation to eating behaviours and food preferences.

## Figures and Tables

**Figure 1 foods-12-01739-f001:**
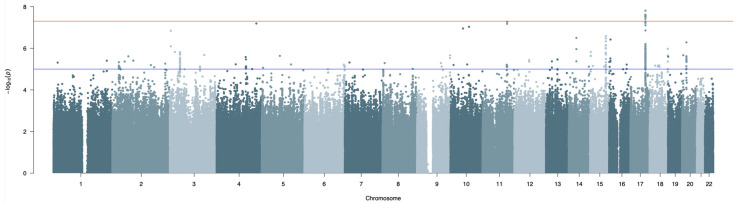
Manhattan plot of GWAS of sweet liking on 1482 individuals from the discovery cohort (Val Borbera and Carlantino). The red line is set at *p*-value = 5 × 10^−8^, and the SNPs above the line were selected for the replication step. The blue line is set at *p*-value = 1 × 10^−5^, and the results for the SNPs above this line are shown in Appendix A. Manhattan plot was generated with the R library qqman [55].

**Figure 2 foods-12-01739-f002:**
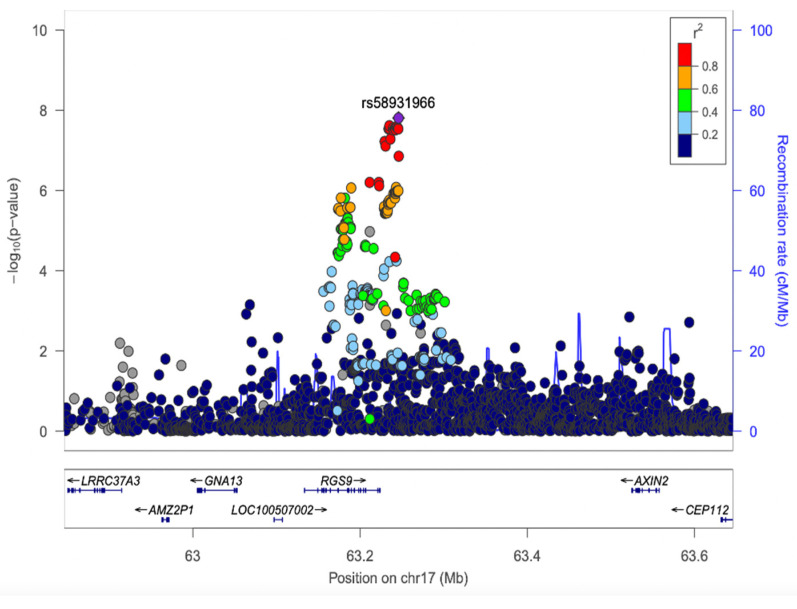
Association plot for the region around rs58931966 in the discovery sample. The purple diamond demonstrates the most strongly associated SNP, rs58931966, near the RGS9 gene. The minus logarithm of single nucleotide polymorphism (SNP) association *p*-value is shown on the *y*-axis and the SNP position (with gene annotation) on the *x*-axis. For each SNP, the strength of LD with the lead SNP is colour-coded by its r^2^. The plot was produced in LocusZoom [56].

**Figure 3 foods-12-01739-f003:**
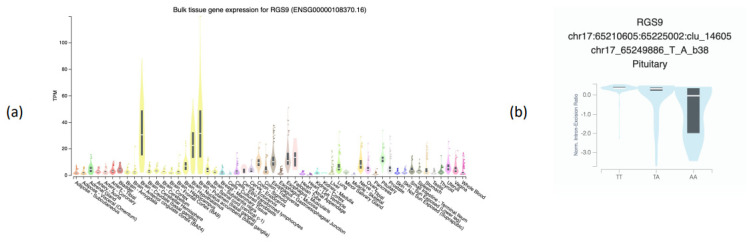
(**a**) Tissue-specific expression (GTEx v8) of *RGS9* gene; (**b**) association of the strongest SNP rs58931966 with the expression of *RGS9* gene on the pituitary tissue (splicing quantitative locus). TMP = transcripts per million.

**Figure 4 foods-12-01739-f004:**
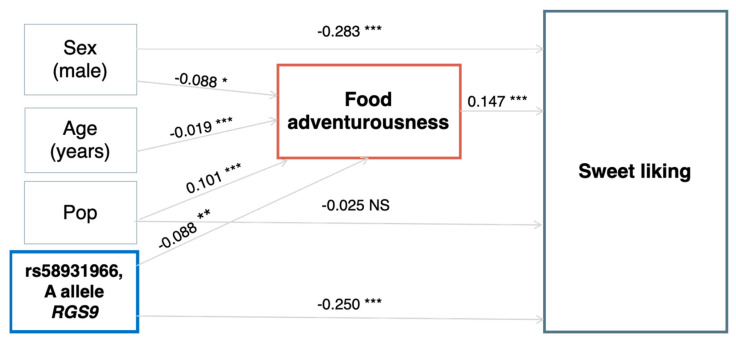
SEM models (Lavaan R package) for sweet liking and food adventurousness (FA). Reported value are βs and *p*-values (* *p*-value < 0.05, ** *p*-value < 0.01, *** *p*-value < 0.001, NS = not significant). Pop = population (Carlantino, Val Borbera, Friuli Venezia Giulia). The model has a good fit (CFI = 0.995, TLI = 0.954, *p*-value Chi-square = 0.094, RMSEA = 0.025). The relationship of the SNP with sweet liking was both direct and mediated by FA (Sobel test, z = 2.04, *p*-value= 0.041).

**Table 1 foods-12-01739-t001:** Characteristics of the participants.

Demographic	Discovery Sample		Replication Sample
	VBI	CAR	FVG
N (M/F)	1109 (434/675)	373 (164/209)	1073 (475/598)
Age, years	56.1 (16.5)	52.4 (17.1)	50.8 (16.2)
Mean liking of sweet foods	6.4 (1.5)	6.4 (1.8)	6.3 (1.6)
FA	1.9 (1.2)	1.8 (1.0)	1.8 (0.9)
RD (*n* = 528)	--	--	52.3 (22.6)
BMI, kg/m^2^	25.6 (4.4)	26.5 (4.8)	25.4 (4.7)
Blood Glucose, mg/dL	89.0 (14.8)	94.9 (16.9)	93.3 (16.2)

When not specified, data are mean and standard deviation (in brackets). BMI = body mass index, CAR = Carlantino, VBI = Val Borbera, FVG = Friuli Venezia Giulia, FA = food adventurousness, RD = reward dependence, -- = no data.

**Table 2 foods-12-01739-t002:** Combined results for the discovery and replication samples for sweet liking.

							Discovery Samples (CAR + VBI)				Discovery and Replication Samples (CAR + VBI + FVG)			
SNP	Chr	Position	Near Gene	Distance	EA/OA	Mean Freq of EA	*n*	Effect	StdErr	*p*-Value	*n*	Effect	StdErr	*p*-Value
rs192789286	11	103927811	*DDI1*	17,889	A/G	0.022	1482	−1.154	0.212	5.32 × 10^−8^	NA	NA	NA	NA
rs9896491	17	63234031	*RGS9*	10,210	A/G	0.760	1482	0.360	0.065	2.90 × 10^−8^	2555	0.286	0.051	1.89 × 10^−8^
rs6504288	17	63234885	*RGS9*	11,064	A/G	0.239	1482	−0.363	0.065	2.85 × 10^−8^	2555	−0.292	0.052	1.40 × 10^−8^
rs6504289	17	63235085	*RGS9*	11,264	A/G	0.760	1482	0.362	0.065	2.42 × 10^−8^	2555	0.285	0.051	2.24 × 10^−8^
rs57333496	17	63236332	*RGS9*	12,511	T/C	0.773	1482	0.359	0.066	5.22 × 10^−8^	2555	0.314	0.053	2.69 × 10^−9^
rs9916255	17	63239497	*RGS9*	15,676	T/C	0.760	1482	0.362	0.065	2.78 × 10^−8^	2555	0.282	0.051	3.19 × 10^−8^
rs55820790	17	63240317	*RGS9*	16,496	A/G	0.774	1482	0.367	0.066	3.13 × 10^−8^	2555	0.321	0.053	1.44 × 10^−9^
rs16961703	17	63240344	*RGS9*	16,523	A/G	0.774	1482	0.367	0.066	3.16 × 10^−8^	2555	0.321	0.053	1.43 × 10^−9^
rs7221051	17	63241713	*RGS9*	17,892	A/G	0.226	1482	−0.368	0.066	2.84 × 10^−8^	2555	−0.322	0.053	1.32 × 10^−9^
rs7221258	17	63241956	*RGS9*	18,135	C/G	0.774	1482	0.369	0.067	2.95 × 10^−8^	2555	0.322	0.053	1.31 × 10^−9^
rs55864812	17	63242632	*RGS9*	18,811	A/G	0.774	1482	0.367	0.066	3.06 × 10^−8^	2555	0.321	0.053	1.38 × 10^−9^
rs7213152	17	63243529	*RGS9*	19,708	T/C	0.774	1482	0.371	0.067	2.56 × 10^−8^	2555	0.323	0.053	1.30 × 10^−9^
rs7212442	17	63243586	*RGS9*	19,765	A/G	0.226	1482	−0.371	0.067	2.56 × 10^−8^	2555	−0.323	0.053	1.30 × 10^−9^
rs16961868	17	63244061	*RGS9*	20,240	A/G	0.775	1482	0.372	0.067	2.57 × 10^−8^	2555	0.324	0.053	1.23 × 10^−9^
rs62063085	17	63245030	*RGS9*	21,209	T/C	0.226	1482	−0.369	0.067	2.75 × 10^−8^	2555	−0.323	0.053	1.25 × 10^−9^
rs7342966	17	63245750	*RGS9*	21,929	T/C	0.774	1482	0.370	0.067	2.91 × 10^−8^	2555	0.323	0.053	1.49 × 10^−9^
rs58931966	17	63246004	*RGS9*	22,183	A/T	0.240	1482	−0.372	0.066	1.56 × 10^−8^	2555	−0.297	0.051	7.05 × 10^−9^

Chr = chromosome, Position is in bp (base pair) build hg19, Distance in bp from the near gene, EA = effect allele, OA = other allele, Mean Freq of EA = mean frequency of effect allele in discovery sample, *n* = number of individuals involved in the analysis, Effect = β of the association, StdErr = standard error of the β. CAR = Carlantino, VBI = Val Borbera, FVG = Friuli Venezia Giulia.

**Table 3 foods-12-01739-t003:** The association of rs58931966 with food adventurousness, reward dependency, body mass index and glucose.

Trait	Population	SNP	Gene	EA/OA	N	Effect	StdErr	*p*-Value
FA	CAR + VBI + FVG	rs58931966	*RGS9*	A/T	2543	−0.065	0.033	0.0490
RD	FVG	rs58931966	*RGS9*	A/T	587	−3.840	1.527	0.011
BMI	CAR + VBI + FVG	rs58931966	*RGS9*	A/T	2519	0.391	0.144	0.007
Glucose	CAR + VBI + FVG	rs58931966	*RGS9*	A/T	2444	1.211	0.489	0.013

BMI = body mass index, EA = effect allele, FA = food adventurous, RD = reward dependence, OA = other allele; CAR = Carlantino, VBI = Val Borbera, FVG = Friuli Venezia Giulia.

**Table 4 foods-12-01739-t004:** Candidate SNPs found associated with sweet liking in discovery plus replication sample (*n* = 2555).

Gene	SNP	Already Associated Phenotype	DOI	Effect Allele	Other Allele	Effect	StdErr	*p*-Value	Direction
*PHACTR2*	rs1416208	Sensory perception of sweet taste	10.1093/AJCN/NQZ043	A	T	0.1392	0.0483	0.00392	+++
*LINC01277*	rs57083985	Sensory perception of sweet taste	10.1093/AJCN/NQZ043	T	G	−0.1317	0.0465	0.004657	−−−
*GRID1*	rs7897266	Aspartame	10.1093/AJCN/NQZ043	T	G	0.2092	0.0825	0.01121	+++
*TAS1R3*	rs35424002	Sweet perception	10.1111/j.1750-3841.2012.02852.x	A	G	0.2284	0.0926	0.01361	−++
*RP11-401I19.2*	rs306356	Intake of total sugars	10.1093/AJCN/NQZ043	A	C	0.2847	0.121	0.01863	−++
*TAS1R2*	rs7534618	Sweet perception	10.1111/j.1750-3841.2012.02852.x	T	G	0.1066	0.0466	0.02214	+++
*ANO3*	rs75941298	Sensory perception of sweet taste	10.1093/AJCN/NQZ043	A	G	−0.172	0.0775	0.02643	−−−
*CAPN13*	rs115354913	Intake of total sugars	10.1093/AJCN/NQZ043	A	G	0.2712	0.1224	0.02677	+++
*SSBP4*	rs8106096	Intake of total sugars	10.1093/AJCN/NQZ043	C	G	−0.104	0.0498	0.03675	−−−
*AC074019.1*	rs62202380	Intake of total sugars	10.1093/AJCN/NQZ043	T	C	−0.2426	0.122	0.04675	+−−

We were able to detect an association between sweet liking and ten already published SNPs. In particular, we replicated the association with an SNP of *TAS1R2* (rs7534618) and one SNP on *TAS1R3* (rs35424002).

## Data Availability

A subset of the data is already available in the European Genome-phenome Archive (EGA) at the following links. FVG cohort: BAM files https://www.ebi.ac.uk/ega/studies/EGAS00001000252 (accessed on 1 December 2020); sample list, vcf files https://www.ebi.ac.uk/ega/studies/EGAS00001001597 (accessed on 1 December 2020); https://www.ebi.ac.uk/ega/datasets/EGAD00001002729 (accessed on 1 December 2020); VBI cohort: BAM files https://www.ebi.ac.uk/ega/studies/EGAS00001000398 (accessed on 1 December 2020); https://www.ebi.ac.uk/ega/studies/EGAS00001000458 (accessed on 1 December 2020); CAR cohort: BAM files https://www.ebi.ac.uk/ega/studies/EGAS00001000460 (accessed on 1 December 2020). A vcf file including all the INGI variants (SNPs and INDELs), with information on allele frequencies in the whole dataset and each cohort, has been submitted to the European Variation Archive (EVA) with study accession number: PRJEB33648. The data are accessible at the following link: https://www.ebi.ac.uk/ena/data/view/PRJEB33648 (accessed on 1 December 2020).

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
