# Peer review of "Regulator of G-Protein Signalling 9: A New Candidate Gene for Sweet Food Liking?"

_foods, 2023, doi:10.3390/foods12091739_

Round 1

Reviewer 1 Report

Thank you for the opportunity to read the interesting material.

At the very beginning, what is thrown are technical errors such as incorrect numbering of chapters. After chapters 1 then comes 4. By the way the methodology is number 1 and the results chapter is also number 1.

The results are very interesting, but there is a balance between the chapters. An entire article cannot be devoted to presenting the results. Complete the content in the remaining sections. In the introduction and discussion chapter, please complete the data and compare it with the data of other researchers, and bibliographies on healthy eating habits. Works by: Kim, S.J.; Åšmiglak-Krajewska, M.; Becerra-Tomas, N.; Viguiliouk, E.; Winham, D.M.

When it comes to bibliographies, more than 50% are items older than 5 years, so supplementing existing sections or adding a "Theoretical Background" section will just increase the relationship between newer literature and classics.

Best Regards,

Author Response

03/04/2023

Manuscript ID: foods-2290705

Dear Editor,                       

We are grateful for your consideration of this manuscript and for the opportunity to submit a further revised version. We have addressed the comments made by the reviewer 1 below. We hope that this version is suitable for publication.

Kind regards,

Catherine Anna-Marie Graham

 Dear Reviewer 1, 

Please find below a point-by-point response to you comments. We would like to thank you for contributing to the strength of our manuscript.

Comment 1

Thank you for the opportunity to read the interesting material. At the very beginning, what is thrown are technical errors such as incorrect numbering of chapters. After chapters 1 then comes 4. By the way the methodology is number 1 and the results chapter is also number 1.

Thank you for highlighting the technical errors. We have corrected the numbering in the manuscript.

Comment 2

The results are very interesting, but there is a balance between the chapters. An entire article cannot be devoted to presenting the results. Complete the content in the remaining sections. In the introduction and discussion chapter, please complete the data and compare it with the data of other researchers, and bibliographies on healthy eating habits. Works by: Kim, S.J.; Åšmiglak-Krajewska, M.; Becerra-Tomas, N.; Viguiliouk, E.; Winham, D.M.

When it comes to bibliographies, more than 50% are items older than 5 years, so supplementing existing sections or adding a "Theoretical Background" section will just increase the relationship between newer literature and classics.

Regarding the introduction, we have included clarification, where relevant, of old versus new publications (line 49, 51, 53-54, 59-61, 68-69, 83, 84-85). We have also included more up to date references, where relevant, throughout; including those suggested by you (line 35-42). We believe this approach is appropriate for our manuscript because a lot of the conclusions from research have remained the same over the past few decades, and additionally, a proportion of the relevant research pertaining to genetics were published a number of years ago. We have ensured to clarify which practices are still used today and where there is need for further research.

Regarding the discussion we have added additional depth around the referenced papers, including a comparison of our results to others throughout. Due to the novel nature of our work, there are not many published papers of direct comparison, however we have ensured to include the most relevant works and theories to place our results into context. We hope that these amendments tackle the feedback provided, and once again would like to thank the Reviewer for their input.

We would like to take this opportunity to thank reviewer 1 for the input on our manuscript, and we hope the changes respond to the comments given.

Kind regards,

Catherine Anna-Marie Graham

Reviewer 2 Report

Dear Authors,

Your study is complex and of great interest for specialists, but the future lecturers of the article should be from variable background and are not always accustomed with those concepts, reason why we made some small suggestions in the hope that it will contribute to be more accessible to a larger public of lecturer interested by nutrition.  

Line 108 you write: “Body Mass Index (BMI) was obtained by Bioelectrical  Impedance Analysis using the Body Composition Analyzer (Tanita BC-420MA; Tanita, Tokyo, Japan).” Body mass index need the height of each subjects, we do not think that bio impedance gives this information. Please correct.

Line 111, please specify where, when, by who the subjects were evaluated.  Say a little bit more on how the blood was collected and prepared for the subsequent analysis.

In your conclusion you write: “If further researched such genetic associations could allow a greater understanding of chronic disease management from both a habitual dietary intake and reward related perspective.” Could you discuss that more clearly in your discussion to help the future lecturers understand how the results of your study can concretely be used for the dietary intake management of chronic disease. Of course, more studies are needed, but what can we concretely do with the results of your study, should be of interest understanding for future lecturers.

Wishes of success!

Author Response

03/04/2023

 Manuscript ID: foods-2290705

Dear Editor,                       

We are grateful for your consideration of this manuscript and for the opportunity to submit a further revised version. We have addressed the comments made by the reviewer 2 below. We hope that this version is suitable for publication.

Kind regards,

Catherine Anna-Marie Graham

Dear Reviewer 2,

Please find below a point-by-point response to your comments. We would like to thank you for your contribution to the strength of our manuscript.

Reviewer #2

Comment 1

Your study is complex and of great interest for specialists, but the future lecturers of the article should be from variable background and are not always accustomed with those concepts, reason why we made some small suggestions in the hope that it will contribute to be more accessible to a larger public of lecturer interested by nutrition.  

Line 108 you write: “Body Mass Index (BMI) was obtained by Bioelectrical Impedance Analysis using the Body Composition Analyzer (Tanita BC-420MA; Tanita, Tokyo, Japan).” Body mass index need the height of each subjects, we do not think that bio impedance gives this information. Please correct.

We apologise for the omission of clear information regarding the anthropometric measurements in our manuscript.  We have now specified height, weight, and BMI measurement techniques as follows (line 122-127):

“Height (m) and weight (kg) were measured and then BMI (kg/m2) was calculated. In VBI and FVG, height was measured to the nearest 0.25 cm using a stadiometer, then weight and BMI were measured using the Body Composition Analyzer (Tanita BC-420MA; Tanita, Tokyo, Japan). In CAR, body weight was measured to the nearest 0.25 kg using a balance-beam scale, and height was measured to the nearest 0.25 cm using a stadiometer, BMI was manually calculated.”

Comment 2

Line 111, please specify where, when, by who the subjects were evaluated.  Say a little bit more on how the blood was collected and prepared for the subsequent analysis.

We would like to thank the reviewer for adding to the specificity of our manuscript.

We have specified where, when, and by who, the participants were evaluated at the beginning of the data collection section (please, see lines 107-111):

Screening sessions were organised in each village in government-provided accommodations. Demographic, lifestyle information and living habits were collected for each participant using a self-administered standard questionnaire. All questionnaires were carried out on the same day, after a detailed explanation by trained staff. Trained personnel were available all the time to answer possible questions of the participants.”

Regarding the blood analysis, we have included further details at lines 129-132:

Fasting blood samples were obtained in separate sessions, in the early morning. Blood was tested the same day or aliquoted and stored for further analysis. Routine biochemical analyses were performed through Cobas 6000 analyzer (Hoffmann-La Roche, Basel, Switzerland).

In addition, we better specified for the DNA analysis at lines 134-135:

Genomic DNA was extracted from blood, using a phenol-chloroform extraction procedure”.

Comment 3

In your conclusion you write: “If further researched such genetic associations could allow a greater understanding of chronic disease management from both a habitual dietary intake and reward related perspective.” Could you discuss that more clearly in your discussion to help the future lecturers understand how the results of your study can concretely be used for the dietary intake management of chronic disease. Of course, more studies are needed, but what can we concretely do with the results of your study, should be of interest understanding for future lecturers.

We added more clarification for future research in the conclusion (line 413 onwards) and have included the theme of chronic disease throughout the manuscript (lines: 35-42, 317-321 and 363-369).

We believe these suggested changes have contributed to the strength of our manuscript, and would therefore like to take this opportunity to thank the reviewer for their comments.

Kind regards,

Catherine Anna-Marie Graham

Round 2

Reviewer 1 Report

Dear authors and where is this NEW if some of the results are the same as a few decades ago?
I appreciate, the attempt to improve the article, but it is done in a bad way. According to the journal's guidelines it should be done in track changes mode to compare what has been added and what has been removed.

Author Response

Dear Reviewer,

We would like to thank you for your contributions to the production of our manuscript. Please find our point-by-point response to your two comments below.

Comment 1

Dear authors and where is this NEW if some of the results are the same as a few decades ago?

I apologise if our previous explanation was unclear; our manuscript presents the novel finding of the Regulator of G-Protein Signalling 9 (RGS9) gene’s association with sweet food preference. To our knowledge this association has not been demonstrated in research before. We state this in line 20-21, 315-319 and 453-454. The research referred to in our previous response to you was in relation to similar studies (GWAS pertaining to food liking); it is these that were published in 2016 (Pirastu et al. 2016) and 2019 (Hwang et al. 2019), both of which are discussed in our manuscript. Regarding the wider context of sweet food liking, which is also highly important to our manuscript, we have included original source references, for example in 1996, Tuorila (1996), whereby patterns of sweet food liking were defined, and much more recent references describing recent opinions on sweet food liking (Armitage et al. 2021). In relation to your previous comment, we have ensured to place the timeframe of the reference into context and state which practices are utilised today (line 49, 51, 53-54, 59-61, 68-69, 83, 84-85).

In addition to the above, we also demonstrate novel findings in relation to the  RGS9 gene, food adventurousness, reward dependency, BMI, and blood glucose. We have taken the same approach regarding references with each topic.

Armitage, R. M.; Iatridi, V.; Yeomans, M. R. Understanding Sweet-Liking Phenotypes and Their Implications for Obesity: Narrative Review and Future Directions. Physiol Behav, 2021, 235. https://doi.org/10.1016/J.PHYSBEH.2021.113398.

Hwang, L. D.; Lin, C.; Gharahkhani, P.; Cuellar-Partida, G.; Ong, J. S.; An, J.; Gordon, S. D.; Zhu, G.; Macgregor, S.; Lawlor, D. A.; et al. New Insight into Human Sweet Taste: A Genome-Wide Association Study of the Perception and Intake of Sweet Substances. Am J Clin Nutr, 2019, 109 (6), 1724–1737. https://doi.org/10.1093/AJCN/NQZ043.

Pirastu, N.; Kooyman, M.; Traglia, M.; Robino, A.; Willems, S. M.; Pistis, G.; Amin, N.; Sala, C.; Karssen, L. C.; van Duijn, C.; et al. A Genome-Wide Association Study in Isolated Populations Reveals New Genes Associated to Common Food Likings. Rev Endocr Metab Disord, 2016, 17 (2), 209–219. https://doi.org/10.1007/S11154-016-9354-3.

Tuorila, H. Hedonic Responses to Falvor and Their Implications for Food Acceptance. Trends in Food Science & Technology , 1996, 7.

Comment 2

I appreciate, the attempt to improve the article, but it is done in a bad way. According to the journal's guidelines it should be done in track changes mode to compare what has been added and what has been removed.

We have amended the yellow highlighting to tracked changes.

Yours Sincerely,

Catherine Anna-Marie Graham